# Modelling the Impact of Reducing Ultra-Processed Foods Based on the NOVA Classification in Australian Women of Reproductive Age

**DOI:** 10.3390/nu14071518

**Published:** 2022-04-05

**Authors:** Nahal Habibi, Shalem Yiner-Lee Leemaqz, Jessica Anne Grieger

**Affiliations:** 1Robinson Research Institute, University of Adelaide, Adelaide, SA 5005, Australia; nahal.habibi@adelaide.edu.au; 2Adelaide Medical School, University of Adelaide, Adelaide, SA 5005, Australia; 3Flinders Health and Medical Research Institute, Flinders University, Adelaide, SA 5042, Australia; shalem.leemaqz@flinders.edu.au

**Keywords:** dietary modelling, simulation modelling, reproductive age, women, ultra-processed food, discretionary nutrients, Australian Health Survey, NOVA classification

## Abstract

Women of reproductive age have a high proportion of overweight/obesity and an overall poor nutritional intake and diet quality. Nutritional modelling is a method to forecast potential changes in nutrition composition that may offer feasible and realistic changes to dietary intake. This study uses simulation modelling to estimate feasible population improvements in dietary profile by reducing ultra-processed food (UPF) consumption in Australian women of reproductive age. The simulation used weighted data from the most recent 2011–2012 National Nutrition and Physical Activity Survey. A total of 2749 women aged 19–50 years was included, and 5740 foods were examined. The highest daily energy, saturated fat, and added sugar and sodium came from UPF. Reducing UPF by 50% decreased energy intake by 22%, and saturated fat, added sugar, sodium, and alcohol by 10–39%. Reducing UPF by 50% and increasing unprocessed or minimally processed foods by 25% led to a lower estimated reduction in energy and greater estimated reductions in saturated fat and sodium. Replacement of 50% UPF with 75% of unprocessed or minimally processed foods led to smaller estimated reductions in energy and nutrients. Our results provide insight as to the potential impact of population reductions in UPF, but also increasing intake of unprocessed or minimally processed foods, which may be the most feasible strategy for improved nutritional intake.

## 1. Introduction

Reproductive life stages include the preconception, pregnancy, and postpartum period, and it typically refers to all women aged 15–49 years [1]. Women in this reproductive age group have demonstrated the greatest rise in the prevalence of obesity [2], with up to 1 kg annual weight gain from early adulthood to middle-age [3]. Whilst such weight gain is known to play an adverse role in maternal and offspring health during pregnancy [4,5], excessive weight gain before, during, and after pregnancy also posits heightened risk for early and future chronic disease risk such as type 2 diabetes and cardiovascular disease [6,7,8].

There is clear evidence indicating that women of reproductive age have poor diet quality and consumption patterns, reflected by a low intake of fruits and vegetables and higher intakes of discretionary foods containing added sugar, sodium, and saturated fat [9,10,11,12,13,14]. Furthermore, there is increasing concern that excess consumption of industrially processed foods is driving the increase in the prevalence of diet-related chronic diseases [15]. Such foods that, for example, include packaged instant soups and noodles, and pre-prepared meat, fish, and vegetables, are often made from cheap ingredients and additives, which are lower in nutritional quality and higher in energy density [16]. While most young and older consumers understand the term ‘ultra-processed’, and they can correctly classify items such as soft drinks, biscuits, and confectionary, they tend to mis-classify some more healthful foods such as milk, flour, meat, cheese, and bread [17,18], potentially contributing to lower intakes of these foods.

Intakes of ultra-processed foods (UPF) are increasing, with food surveys demonstrating around 27–60% total daily energy intake in adults from UPF [5,19,20,21], including 38.9% of total energy intake among Australian adults [22]. Limited studies have specifically examined UPF intake in women; intake of UPF was 33% in UK women aged 19–60 years [23], 51.2% of total daily energy intake in Brazilian women aged 21 to 24 years of age [24], and 59% in Korean women aged 19–64 years [25]. High intakes of UPF, equivalent to approximately 76% of total energy intake is associated with a 30% higher risk for obesity in Canadian adults [21], and in an adult French population a 10% increase in the proportion of UPF is associated with an approximate 10% higher risk for cardiovascular disease [26] and cancer [27]. In Australia, an increased dietary share of UPF was also associated with higher BMI [22], and an increased risk of inadequate intakes of nutrients critical to obesity and other non-communicable diseases [28].

Modelling studies use analytical methods that account for events over time and across populations and that are based on data drawn from a range of sources [29]. Within the Australian context, nutritional modelling has been used to forecast or predict changes in dietary intake quantity and/or composition that are needed to achieve certain targets, such as the Australian Dietary Guidelines [30], and that has demonstrated impactful population level reductions in salt and trans fatty acid intake, along with implementation of sodium and trans-fat reformulation programs [31]. Modelling taxes on saturated fat, salt, sugar, and sugar-sweetened beverages, and placing a subsidy on fruits and vegetables, estimated cost-savings for the Australian health sector [32], with more than 30,000 deaths from coronary heart disease, stroke and cancer predicted to be saved if UK dietary guidelines were met [33]. No studies have assessed the impact of theoretical changes to UPF consumption using Australian data. This is clearly of relevance, particularly in women of reproductive age, given their overall poor dietary quality, increasing rates of overweight and obesity, and that diet and lifestyle intervention studies are currently ineffective for consistent weight management or loss, particularly during pregnancy and post-partum [34]. In reproductive age women participating in the 2011–2013 Australian Health Survey, the aims of this study are to (1) describe the energy, macronutrient, and discretionary nutrient profile according to the NOVA food classification and their contribution to total daily energy intake; and (2) use simulation modelling to estimate feasible population improvements in dietary profile by reducing ultra-processed and processed food consumption. Outcomes from the modelling scenarios will enable the development of dietary interventions to improve diet quality and support body weight loss in these at-risk women.

## 2. Materials and Methods

### 2.1. Study Population

The data source for this study was the National Nutrition and Physical Activity Survey (NNPAS) 2011–2012, part of the 2011–2013 Australian Health Survey [35]. This survey studied a randomly selected, national sample (*n* = 12,153) of the Australian population using a complex, stratified, multistage probability cluster sampling design with selection of strata, households, and people within households. The current analysis used the food intake data from the first 24-h recall among reproductive aged women 19–50 years (*n* = 2749), and population weighted using sample weighting factors provided in the survey [35]. The Census and Statistics Act, 1905, provided the Australian Bureau of Statistics with the authority to conduct NNPAS, with all respondents providing written informed consent.

### 2.2. Dietary Data

AUSNUT 2011–2013 was the Food Standards Australian and New Zealand nutrient database [36] developed to enable food, dietary, supplement, and nutrient intake estimates to be made from the 2011–2013 Australian Health Survey [35]. The AUSNUT 2011–2013 database groups foods according to a major (2-digit), sub-major (3-digit), or minor (5-digit) food group. The 5-digit group then forms the basis of the survey ID (8-digit) assigned to each food, beverage, or ingredient, as previously described [37]. Weighted population averages for each unique 8-digit food code were obtained using SPSS software (version 25, IBM SPSS Inc., Chicago, IL, United States). Population intakes of food (grams), energy, macronutrients, and discretionary components (mean intake/day) were aggregated with the food data and nutrient values (per 100 g) from AUSNUT 2011–2013 [36] in Microsoft Excel (2019, Microsoft Corporation, Redmond, WA, USA).

Each food was allocated to one of the four NOVA food groups [38]. The NOVA system is a food classification based on the nature, extent, and purpose of industrial food processing, which classifies foods into four groups: unprocessed and minimally processed foods; processed culinary ingredients; processed foods; and ultra-processed foods (UPF) [39]. The NOVA food classification system was based on what was previously applied to all 5740 8-digit food and beverage items in AUSNUT 2011–2013 [37,40]. As such, results presented in the simulation models and the NOVA foods are slightly different from each other, as the NOVA system classifies each individual food item within a mixed food item to a NOVA group and is more granular, whereas in the modelling dataset, the entire food is allocated to either an unprocessed, processed, or minimally processed food group.

### 2.3. Dietary Scenarios

The first model strategy used a simulation model to reduce the gram weight of all UPF by 50% (Model 1). This strategy was chosen to demonstrate the effects of reducing predominantly discretionary/unhealthy food choices in the diet that are high in saturated fat, added sugars, and sodium, and that form around twice the recommended intake of discretionary choices in the Australian diet [41]. Whilst a simple reduction in UPF may appear a feasible option to reduce energy intake and discretionary nutrient profile, two supplementary strategies were tested; that is, in conjunction with reducing the weight of all UPF by 50% (Strategy 1, Model 1), the quantities of unprocessed or minimally processed foods were increased by 25% (Strategy 1, Model 2) and 75% (Strategy 1, Model 3). Unprocessed or minimally processed foods are typically fresh fruits and vegetables, grains (cereals), pasteurized full fat, low-fat, skimmed milk, and fermented milks, and meats, poultry, fish, and seafood. The reported intakes of these foods are typically suboptimal [28,41]; thus, it is important to examine the estimated impact of increasing intake of unprocessed and minimally processed foods with a concomitant reduction in UPF.

The second strategy was a simulation model to reduce all processed foods, which are typically processed meat and fish such as ham, bacon, and dried fish; cheeses made from milk, salt, and ferments; and unpackaged freshly made breads; and beer, cider, wine by 50%. The same strategies that were used for modelling UPF were also modelled for processed foods, that is, a simulation model to reduce all processed foods by 50% (Strategy 2, Model 1); and in conjunction with reducing all processed foods by 50%, intake of unprocessed and minimally processed foods were increased by 25% (Strategy 2, Model 2) and 75% (Strategy 2, Model 3).

### 2.4. Dietary Modelling

To investigate the impact of the different modelling scenarios below, the Microsoft Excel Solver add-in was used to manipulate baseline food and beverage quantity (grams). Nutrition modelling was undertaken at the population level, such that food group intakes were aggregated at the 8-digit food level (*n* = 4028 foods and beverages).

## 3. Results

### 3.1. Population Baseline Intakes

Women aged 19–50 years from the Australian Health Survey were included (*n* = 2749) and population weighted. Population mean daily intakes of energy, macronutrients, and key discretionary nutrients are reported in Table 1. Women consumed a mean 3.1 kg/d food, totalling a mean energy intake of 7388 kJ/d (1765 kcal/d). The percent energy from protein, carbohydrate, fat, saturated fat, and added sugars was a respective 16.9%, 45.0%, 32.6%, 12.4%, and 10.5%, with respective mean daily intakes of fibre and alcohol 19.8 g/d and 9.1 g/d.

Ultra-processed foods contributed 41.4% of total daily energy intake, with the included foods contributing to one fifth of daily grams of food consumed (Table 1). Women consumed the highest percentage of energy, total fat, carbohydrate, saturated fat, and added sugars and sodium from UPF, whereas the highest percentage of protein and fibre was from unprocessed or minimally processed food. The largest gram weight of food came from unprocessed or minimally processed foods, contributing around a third of total daily energy intake, and more than half of total protein and fibre intake. Processed foods contributed the highest amount of alcohol, 7.3 g/d, contributing 14.8% of energy, and 21.6% of sodium to the diet.

Table 2 shows the contribution of different foods and drinks to mean daily energy intake. Mass-produced packaged breads, pastries, buns, and cakes and fast foods dishes contributed the highest energy from UPF. Processed breads and beer and wine accounted for about half of the energy contribution from processed foods. Within the unprocessed or minimally processed foods category, around 6–8% percent of energy came from red meat and poultry, cereal grains and flours, and milk and plain yoghurt. Plant oil had the highest contribution to daily energy within the processed culinary ingredients group.

### 3.2. Strategy 1

#### 3.2.1. Model 1: Reducing Ultra-Processed Foods by 50%

The impact of reducing UPF by 50% is shown in Figure 1 (green bars). Compared to population baseline intakes, halving the intake of UPF resulted in a theoretical 316 g lower intake of total food consumed, and a 1689 kJ (404 kcal) lower daily energy intake. Modelled intakes were lower for all discretionary components such as saturated fat, added sugar, and sodium by 20–40%, with a lower intake of alcohol by 10.0%. Halving the intake of UPF also reduced macronutrients by 15–30%.

#### 3.2.2. Models 2: Reducing Ultra-Processed Foods by 50% and Increasing Unprocessed or Minimally Processed Foods by 25%

Figure 1 shows the impact of replacing 50% of UPF with 25% (blue bars) or 75% (orange bars) of unprocessed or minimally processed foods. Compared with the baseline intake, partial replacement by 25% reduced daily energy intake by 1131 kJ (270 kcal), while the quantity of all consumed foods increased by 231 g. This model also led to theoretical reductions in saturated fat (15.1%), added sugars (38.3%), sodium (23.4%), and alcohol (10.0%). Modelled food intakes were lower in protein, fat, carbohydrate, and fibre by 5.6%, 13.1%, 20.2%, and 8.4%, respectively.

#### 3.2.3. Model 3: Reducing Ultra-Processed Foods by 50% and Increasing Unprocessed or Minimally Processed Foods by 75%

Partial replacement of UPF with 75% unprocessed or minimally processed foods resulted in a hypothetical increase in the quantity of food consumed by around 1.3 kg (Figure 1, orange bars). Energy intake was not shown to change (−16 kJ/d [3.8 kcal]), whereas protein and fibre increased (12–15%), and carbohydrate intake decreased (6.3%). Modelled intakes led to a reduction in all discretionary components, with the largest decrease in intakes of added sugars.

### 3.3. Strategy 2

#### 3.3.1. Model 1: Reducing Processed Foods by 50%

Figure 2 (green bars) shows the theoretical changes in food consumed, energy, macronutrients, and discretionary nutrients by reducing processed foods by 50%. Compared to population baseline intakes, reducing processed foods reduced the intake of grams of food consumed (165 g), daily energy intake (913 kJ), and macronutrients (7–14%). Discretionary nutrients were reduced by 4–14%, except for alcohol with an estimated reduction of 40% (−3.6 g).

#### 3.3.2. Model 2: Reducing Processed Foods by 50% and Increasing Unprocessed or Minimally Processed Foods by 25%

Replacing 50% of processed foods with 25% of unprocessed or minimally processed foods led to estimated reductions in energy intake of 355 kJ (85 kcal), protein, fat, and carbohydrates by 0.4% to 7.6%, but an estimated 0.7% increase in fibre intake (Figure 2, blue bars). The estimated reduction in alcohol (40%) was the same to when there was no increase intake of unprocessed or minimally processed foods (Model 1), and the reduction in discretionary nutrients was similar (3.8% to 10.2%).

#### 3.3.3. Model 3: Reducing Processed Foods by 50% and Increasing Unprocessed or Minimally Processed Foods by 75%

The impact of replacing 50% of processed foods with 75% of unprocessed or minimally processed foods is shown in Figure 2 (orange bars). The model led to a theoretical 1.5 kg higher intake of food consumed and 760 kJ (182 kcal) increase in energy intake in comparison with population baseline intakes. Macronutrients were estimated to increase by 6–17% and fibre by 4.5 g. The estimated reduction in alcohol remained the same at 40%, whether unprocessed or minimally processed foods were increased or not, but there were smaller estimated reductions in added sugars (2.9%), and sodium (2.3%), but an increase in saturated fat of 4.1% (1 g).

## 4. Discussion

This study describes the NOVA food classification and potential reductions in energy, macronutrients, and discretionary nutrients, following simulation modelling of processed and unprocessed foods. We extend our previous work demonstrating the overall low diet quality in reproductive age women from the same Australian Health survey [42], to now report that a high proportion of energy intake comes from UPF. Our modelling shows that halving intake of processed foods resulted in an estimated reduction in energy (11.9%), saturated fat (14.3%), added sugar (4.3%), sodium (14.1%), and alcohol (40.0%), whereas halving UPF resulted in a greater reduction in energy (22.0%) and discretionary components, namely saturated fat, added sugar, and sodium (21–39%) but not alcohol (10.0%).

The highest amount of total daily energy intake (41.4%, or 3 MJ of energy) was from UPF. Using the Australian Health Survey, older children and adolescents were reported to consume around 54% of total daily energy intake from UPF [28], and within the 2011–2013 Australian food composition database, the proportion of UPF was 38%, and unprocessed and minimally processed food was 36% [37]. Whilst many foods are incorporated into the UPF category, the fact that the contribution to energy intake is higher than that from unprocessed or minimally processed foods, at 36% (2.7 MJ), is concerning, given their link to obesity and non-communicable diseases [43]. Such high consumption may be partly due to their longer shelf life than fresh foods, affordability, appetizing and palatable qualities, and less effort and time required for preparation and cooking [44,45,46]. Furthermore, featuring inaccurate nutrition and health statements on UPF packaging may make them appear healthier than they really are [44,45,46]. Improved accuracy of food labels would allow consumers to better understand the nutritional content of foods and select more nutritious foods. While the Australian dietary guidelines do not currently incorporate NOVA food categories but instead describe discretionary choices [47], future and/or alternative guidelines may be warranted with a focus on level of processing.

The first simulation strategy, reducing UPF by 50%, demonstrated the largest theoretical reductions for all macronutrients as well as saturated fatty acids and sodium; however, reductions in added sugar and alcohol were similar in the energy compensation scenarios that included a 25% or 75% increase in unprocessed or minimally processed foods. Reducing UPF intake led to a proposed reduction in energy (~1700 kJ; 400 kcal), which would be equivalent to the energy of nearly three servings of discretionary choices [47], or several servings of foods within this category. For sustained weight loss, a continued energy restriction of between 2000 and 3000 kJ (500–750 kcal) is recommended [48]. However, it is evident that the practicality of reducing energy intake is rarely achieved over the longer term [49], and continuous energy restriction may be problematic, partly due to increases in the desire to eat [50], feelings of hunger [51], and cravings [52]. Thus, while reducing intakes of UPF offers a potential strategy with important reductions in discretionary nutrients, and which would contribute to improvements in lipid profile [53], the long term success of this strategy is likely hampered by other genetic [54], behavioural, and hormonal [55,56] factors that mitigate sustained weight loss efforts.

The most practical solution from this strategy is likely to be reducing UPF by 50% and increasing unprocessed or minimally processed foods by 25%. A smaller, albeit important, reduction in energy was apparent, along with considerable reductions of added sugar (17.8 g, ~4.5 teaspoons) and saturated fat (3.8 g, 15.1%). To implement this, one could eliminate 100 g French fries from the diet, eliminate 375 mL soft drink and two sweet biscuits, or reduce intake of bread and sweet pastries/buns by half. These foods could be replaced with any two of the following examples: ½ cup vegetables, ½ serving meat or nuts; one serving of fruit. This strategy could assist in weight management programs and potential future risk reduction of chronic diseases such as obesity and type 2 diabetes. A previous study in adults aged over 18 years, also using Australian Health Survey data, highlighted that substituting unhealthier/discretionary foods with a range of healthier foods lowered intakes of added sugars, sodium, and saturated fat and appeared the most feasible strategy for improving nutritional intake [57].

Processed foods contributed nearly 15% of total daily energy intake, of which processed breads contributed 5.8% and beer and wine contributed 3.2%. Reducing processed foods by 50% resulted in around half the reduction in energy from what was observed when UPF was reduced by 50%. Importantly, in this scenario there was a theoretical reduction in alcohol of 40% (3.6 g), or equivalent to just under a half of a standard alcoholic drink. Whilst this may seem small, at a population level, this has huge implications for pursuing a healthier lifestyle and lowering the burden on health services. There has been a continued increase in alcohol consumption among women of reproductive age, not only in Australia [58], but also in the USA [38] and worldwide [39]. This is a critical issue for women who are intending to become pregnant, as higher intakes of alcohol in the preconception period is associated with a longer time to conceive [59]. Similar to the first modelling strategy, incorporating 75% of unprocessed or minimally processed foods is likely to be the least feasible. A 1.5 kg increase in food volume was predicted, and although beneficial increases in protein and fibre were estimated, there were only small changes to discretionary nutrients. The potential benefits of reducing processed foods and increasing by 25% unprocessed and minimally professed foods is unclear. While a modest reduction in daily energy intake of 355 kJ was predicted, which could be helpful for women who opt for weight maintenance, the changes to discretionary nutrients were not remarkable, and the longer-term health outcomes of such a change is difficult to establish. For women who do not consume alcohol, benefits to reducing intake of processed foods, particularly lowering total fat, saturated fat, and sodium would be apparent; however, future studies could investigate such strategies in sub-groups of women.

Our study extends previous findings using the NOVA system, and which could be applied to intervention studies in women of reproductive age. The high proportion of energy from UPF that we report, and which is consistent with previous studies, further compels public health strategies to be developed to monitor the accuracy of food labels and inform women about nutrition risks of consuming these foods, but also the potential adverse reproductive health outcomes. Related, more than 8000 packaged foods in Australia bear a Health Star Rating, a nutrient-based front-of-pack labelling scheme [60]. Three-quarters of UPF display a Health Star Rating quality of ≥2.5, which is a ‘pass’ rating [61]. Thus, discerning between UPF and foods that are marketed as ‘healthy’ is another challenge. Women continue to be the regular supermarket shoppers, they more often shop with children who have a key influence on household purchasing behaviour, but they also shop hurriedly [62]. As such, there is little time to digest and interpret front of pack labels, and to make informed decisions. Globally, the availability of UPF is high. Trends in the purchase and sales of UPF demonstrate the greatest consumption in high-income countries, but they are increasing in lower- and middle-income countries [16]. Interestingly, Australian data show that the lowest household income quintile consumes less UPF [63]. However, this may reflect healthy diets being cheaper than non-healthy diets in general in Australia [64,65]; the cost of UPF vs. non-UPF is not currently available. Thus, while the proposed dietary scenarios may impact the family food cost, future research on the potential impact on food prices will be helpful. Lastly, lower socio-economic status is consistently associated with higher UPF consumption [19,63,66]. The lack of access to fresh foods and the predominance of convenience foods, particularly in low-resource settings [67,68], presents an ongoing challenge and complexity of the relationship between improving diet quality and decreasing UPF consumption. A multi system approach is clearly needed that supports improved knowledge on, and encourages the consumption of, unprocessed and minimally processed foods; minimizes structural barriers such as access to healthy foods; improves their affordability; and attends to minimizing gender inequalities and food insecurity.

This is the first study to explore the potential impact of reducing UPF in Australian diets. We developed the strategies based on the unmet need to improve diet quality in reproductive aged women, and to identify a potential feasible strategy that could be applied in an intervention setting. We considered various simulation models, including different options to not only reduce unhealthy food choices, but to allow for energy compensation through increasing healthier food choices. The use of systematically collected nutrition data in the Survey provides a convincing effect of the potential impact on energy and nutrient intake. A strength of this study was the use of individual-level dietary survey data taken from a nationally representative sample of Australian children and adults. However, the modelling conducted was only in women aged 19–50 years, thus reducing the generalizability of our findings, and whether such modelling outcomes differ across different regions and socio-demographics is unclear. Limitations also include the use of a single 24-h dietary recall and thus may not reflect usual intake and potential misclassification of the level of food processing from food composition databases [38]. Our modelling assumes that all reproductive age women will make the changes to their diet, but this is an unrealistic expectation. The models that include allowances for extra unprocessed and minimally processed foods account for some of this difficulty. Furthermore, the NOVA classification categorizes foods according to food processing and not nutritional content. The food composition database is not designed to categorize foods in this way, thus errors related to the classification of foods cannot be excluded. Finally, while we used the most recent population nutrition survey data, this was reported on 10 years ago, and food consumption patterns would have likely changed since then.

## 5. Conclusions

In conclusion, Australian women of reproductive age consume nearly half of the energy in their diets from UPF. Reducing UPF by 50% considerably lowers estimated energy intake and discretionary nutrients; however, incorporation of 25% of unprocessed or minimally processed foods may be the most feasible strategy for improved health over the longer term. Reducing processed foods offers an important health strategy, particularly for women who consume alcohol; however, the relevance to women who do not drink requires further investigation. Study results can contribute to the development of dietary interventions to improve health, including potential weight loss that utilizes a multi system approach that encourages increased education and behaviour change strategies.

## Figures and Tables

**Figure 1 nutrients-14-01518-f001:**
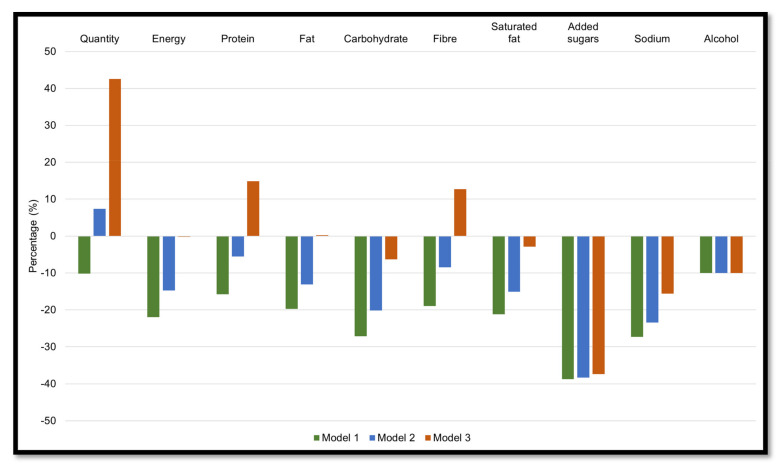
Estimated changes in population mean intakes of food (g), energy, macronutrients, and discretionary nutrients, according to Strategy 1 (3 different models). Results are presented as a percentage change relative to baseline intake. Model 1: Reducing ultra-processed foods by 50%. Model 2: Reducing ultra-processed foods by 50% with a 25% increase in unprocessed or minimally processed foods. Model 3: Reducing ultra-processed foods by 50% with a 75% increase in unprocessed or minimally processed foods.

**Figure 2 nutrients-14-01518-f002:**
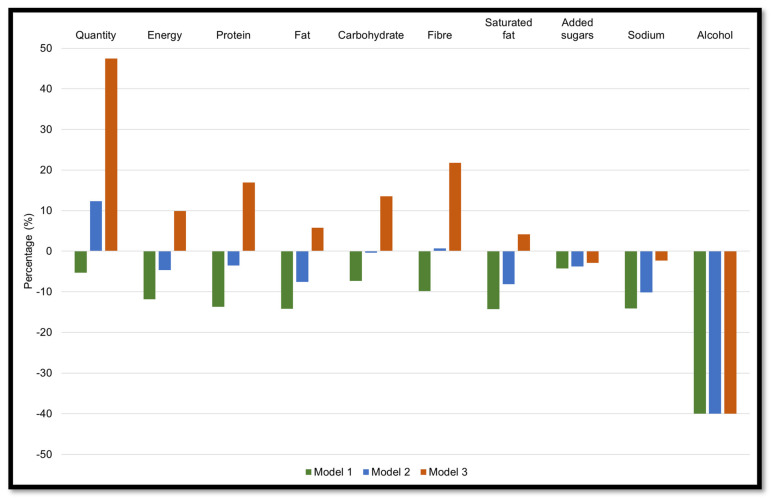
Estimated changes in population mean intakes of food (g), energy, macronutrients, and discretionary nutrients, according to Strategy 2 (3 different models). Results are presented as a percentage change relative to baseline intake. Model 1: Reducing processed foods by 50%. Model 2: Reducing processed foods by 50% with a 25% increase in unprocessed or minimally processed foods. Model 3: Reducing processed foods by 50% with a 75% increase in unprocessed or minimally processed foods.

**Table 1 nutrients-14-01518-t001:** Mean population baseline intakes of Australian women aged 19–50 years, and according to the NOVA classification (*n* = 2749).

	Mean Baseline Intake	Intake from Ultra-Processed Foods (%)	Intake from Processed Foods (%)	Intake from Unprocessed or Minimally Processed Foods (%)	Intake from Processed Culinary Ingredients (%)
Quantity (g)	3112.6	548.9 (17.6)	175.3 (5.6)	2361.8 (75.9)	26.6 (0.9)
Energy (kJ) *	7388.2	3056.2 (41.4)	1093.0 (14.8)	2656.1 (36.0)	582.9 (7.9)
Protein (g)	74.6	20.8 (27.9)	10.9 (14.6)	42.7 (57.2)	0.2 (0.3)
Fat (g)	63.9	26.5 (41.5)	8.7 (13.6)	17.1 (26.8)	11.6 (18.2)
Carbohydrate (g)	198.7	97.1 (48.9)	20.8 (10.5)	71.5 (36.0)	9.3 (4.7)
Fibre (g)	19.8	6.3 (31.8)	2.4 (12.1)	11.1 (56.1)	0.0 (0.0)
Saturated Fat (g)	24.3	10.6 (43.6)	4.0 (16.5)	6.0 (24.7)	3.7 (15.2)
Added sugar (g)	46.2	36.8 (79.7)	1.4 (3.0)	0.2 (0.3)	7.9 (17.1)
Sodium (mg)	2142.3	1309.5 (61.1)	463.4 (21.6)	289.8 (13.5)	79.6 (3.7)
Alcohol (g)	9.1	1.8 (19.8)	7.3 (80.2)	0.0 (0.0)	0 (0.0)

* To convert kJ to kcal, divide by 4.1868.

**Table 2 nutrients-14-01518-t002:** Mean absolute and relative daily energy intake of Australian women aged 19–50 years, according to the NOVA food classification (*n* = 2749).

NOVA Food Groups	Energy (kJ)	Energy (kcal)	% of Total Energy Intake
**Ultra-processed foods**	3056.2	730.4	41.4
Mass-produced packaged breads	333.5	79.7	4.5
Pastries, buns, and cakes	292.3	69.9	4.0
Fast foods dishes ^a^	286.7	68.5	3.9
Confectionery	247.9	59.2	3.4
Frozen and shelf stable ready meals ^b^	237.4	56.7	3.2
Fruit drinks and iced teas	206.2	49.3	2.8
Breakfast cereals	190.4	45.5	2.6
Biscuits	180.4	43.1	2.4
Carbonated soft drinks	171.7	41.0	2.3
Milk-based drinks	168.5	40.3	2.3
Sausage and other reconstituted meat products	163.5	39.1	2.2
Sauces, dressing, and gravies	157.7	37.7	2.1
Salty snacks	118.5	28.3	1.6
Ice cream, ice pops, and frozen yoghurts	101.7	24.3	1.4
Margarine and other spreads	91.3	21.8	1.2
Alcoholic distilled drinks	53.7	12.8	0.7
Other ^c^	54.8	13.1	0.7
**Processed foods**	1093.0	261.2	14.8
Processed breads	427.0	102.1	5.8
Beer and wine	233.5	55.8	3.2
Cheese	220.2	52.6	3.0
Bacon and other salted, smoked, or canned meat or fish	84.0	20.1	1.1
Vegetables and other plant foods preserved in brine	36.2	8.7	0.5
Other ^d^	92.1	22.0	1.2
**Unprocessed or minimally processed foods**	2656.1	634.8	36.0
Red meat and poultry	582.5	139.2	7.9
Cereal grains and flours	485.8	116.1	6.6
Milk and plain yoghurt	452.7	108.2	6.1
Fruits ^e^	323.2	77.2	4.4
Vegetables	239.2	57.2	3.2
Pasta	204.8	48.9	2.8
Nuts and seeds	96.3	23.0	1.3
Potatoes and other tubers and roots	80.5	19.2	1.1
Eggs	71.7	17.1	1.0
Fish	62.1	14.8	0.8
Legumes	31.7	7.6	0.4
Other ^f^	25.8	6.2	0.3
**Processed culinary ingredients**	582.9	139.3	7.9
Plant oils	269.6	64.4	3.6
Animal fats	164.3	39.3	2.2
Table sugar	125.6	30.0	1.7
Other ^g^	23.4	5.6	0.3
Total	7388.2	1765.8	100.0

^a^ Hamburger, pizza, and French fries from fast food places; ^b^ frozen lasagne, pizza, and other pastas and meals, and instant soups and noodles; ^c^ ultra-processed cheese, baby food, and baby formula; ^d^ salted or sugared nuts, seeds, and dried fruits; ^e^ fruits and freshly squeezed juices; ^f^ meat from other animals, teas, coffees, and dried spices; ^g^ honey, maple syrup (100%), and vinegar.

## Data Availability

Data and material are available for data transparency.

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
