# Peer review of "Modelling the Impact of Reducing Ultra-Processed Foods Based on the NOVA Classification in Australian Women of Reproductive Age"

_nutrients, 2022, doi:10.3390/nu14071518_

Round 1
Reviewer 1 Report
Thank you the preparation of this well presented manuscript. Ultra-processed foods are currently very topical and the analysis based on the NOVA classification is useful. The major drawback of the study is that the survey data is 10 years old so any more recent investigations would be helpful to add to the discussion.
The tables provide mean intakes of Australian women aged 19-50 years. Did you consider how intake of UPF might differ by age group? Studies 25 and 26 in reference to young women appear to be outside of Australia. Is there any evidence from national surveys? This may be important for targeted intervention strategies given the influence of marketing on the consumption of UPF. The pattern of sociodemographic factors and health behaviours that are linked with dietary intakes of UPF is an important consideration. For example, reference 11 suggests pregnant women have a higher fruit intake.
Please consider the following reference -Machado et al. Ultra-processed food consumption and obesity in the Australian adult population Nutrition and Diabetes (2020) 10:39 https://doi.org/10.1038/s41387-020-00141-0
This is also a cross-sectional analysis of dietary data from 7411 Australians aged ≥20 years from the National Nutrition and Physical Activity Survey 2011–2012 and where NOVA system was used to identify UPF. Notably those in the highest quintile of UPF food consumption had significantly higher BMI.
The following suggestions are provided to provide clarification to the reader.
Introduction
Line 50 Please add context for the studies 25-26 and 20-22 this is important if outside Australia.
Line 54 Refer the reader again to the Australian survey
Lines 58-60 Again provide the context for studies 23,24. Were they in reproductive women? What geographical area?
Lines 67-70 Is this modelling for the Australian context, if so please say so.
Lines 71-73. This sentence is a repeat of paragraph one, please rewrite this section.
Discussion.
Limitations
Please consider the modelling of all reproductive age women as a limitation. How might the modelling differ across socio-demographic factors?
Please address as a limitation the current analysis used the food intake data from the first 24-hr recall of the survey. Consider that analyses based on a single recall and may not represent the usual diet.
Consider the NOVA system is a food classification based on the nature, extent, and purpose of industrial food processing but the food composition
database is not designed to categorise foods in this way so errors related to classification of foods cannot be excluded.
Line 322 Please describe why it is problematic that >8000 foods have a health star rating?
Line 334 Yes agree a multi system approach is clearly needed. Furthermore UPF are a dominant component of the food supply and this raises environmental concerns as these come at the expense of the cultivation, manufacture and consumption of traditional minimally processed foods.
Author Response
Dear Editor and Reviewers,
We thankyou very much for the kind and constructive feedback that has been provided on our manuscript (nutrients-1672887). We have amended our manuscript using track changes, and we have made a point-by-point response to each of the comments below. We have also uploaded a graphical abstract.
Thankyou for considering our manuscript.
Jessica Grieger
Comment 1: Thank you the preparation of this well presented manuscript. Ultra-processed foods are currently very topical and the analysis based on the NOVA classification is useful. The major drawback of the study is that the survey data is 10 years old so any more recent investigations would be helpful to add to the discussion.
Response 1: Thankyou for your positive feedback. We agree that recent data would be useful, particularly given dietary changes that have likely occurred over this time. Unfortunately, the data we have used is the most recent National Health Survey that collected dietary intake data and we look forward to more recent national Australian surveys to assess current consumption patterns and changes over time.
Comment 2: The tables provide mean intakes of Australian women aged 19-50 years. Did you consider how intake of UPF might differ by age group? Studies 25 and 26 in reference to young women appear to be outside of Australia. Is there any evidence from national surveys? This may be important for targeted intervention strategies given the influence of marketing on the consumption of UPF. The pattern of sociodemographic factors and health behaviours that are linked with dietary intakes of UPF is an important consideration. For example, reference 11 suggests pregnant women have a higher fruit intake.
Response 2: We recently published the dietary intakes of Australian reproductive age women, comparing women aged 19-35 years to those age 35-50 years olds, using the same health survey, which we now reference in this study (Habibi et al, 2021, Nutrients). We found no difference between older and younger women in meeting food group recommendations, with most having poor diet quality overall, a small proportion meeting fruit and veg recommendations, and with no difference in intakes whether women had or did not have children. Given this, we did not split the data by age group for the purpose of nutritional modelling, and because we were looking at reproductive age women as a whole. The sample size of pregnant women in the Australian Health Survey was also small (<200 women) and we could not make a comparison to those women. We acknowledge that food intake data in pregnant women is limited and inconsistent between studies, regarding whether they consume healthier diets compared to non-pregnant women. We had reported on food survey data in adults (references 19-22), but data in women and specifically between different age groups of women is limited.
In terms of patterns of sociodemographic factors, we have now made a statement regarding the income and socioeconomic link in both the discussion and limitations sections on page 6: “Interestingly, Australian data show that the lowest household income quintile consumes less UPF [65]. However, this may reflect healthy diets being cheaper than non-healthy diets in general in Australia [66,67]; the cost of UPF vs non-UPF is not currently available. Lastly, lower socio-economic status is consistently associated with higher UPF consumption [19,65,68].”
“However, the modelling conducted was only in women aged 19-50 years, thus reducing the generalizability of our findings, and whether such modelling outcomes differ across different regions and socio-demographics is unclear.”
Comment 3: Please consider the following reference -Machado et al. Ultra-processed food consumption and obesity in the Australian adult population Nutrition and Diabetes (2020) 10:39 https://doi.org/10.1038/s41387-020-00141-0
This is also a cross-sectional analysis of dietary data from 7411 Australians aged ≥20 years from the National Nutrition and Physical Activity Survey 2011–2012 and where NOVA system was used to identify UPF. Notably those in the highest quintile of UPF food consumption had significantly higher BMI.
Response 3: Thankyou, we have included this reference in the introduction to report both on the % UPF in the Australian adult population and the association to increasing BMI.
Comment 4: The following suggestions are provided to provide clarification to the reader.
Introduction
Line 50 Please add context for the studies 25-26 and 20-22 this is important if outside Australia.
Line 54 Refer the reader again to the Australian survey.
Lines 58-60 Again provide the context for studies 23,24. Were they in reproductive women? What geographical area?
Lines 67-70 Is this modelling for the Australian context, if so please say so.
Lines 71-73. This sentence is a repeat of paragraph one, please rewrite this section.
Response 4: We have attended to these and added in county and age groups where appropriate.
Discussion.
Comment 5:
Limitations
Please consider the modelling of all reproductive age women as a limitation. How might the modelling differ across socio-demographic factors?
Response 5: We have now added to the strengths/limitations section on page 6: “A strength of this study was the use of individual-level dietary survey data taken from a nationally representative sample of Australian children and adults. However, the modelling conducted was only in women aged 19-50 years, thus reducing the generalisability of our findings, and whether such modelling outcomes differ across different regions and socio-demographics is unclear.”
Comment 6: Please address as a limitation the current analysis used the food intake data from the first 24-hr recall of the survey. Consider that analyses based on a single recall and may not represent the usual diet.
Response 6: We now state “Limitations include the use of a single 24-hr dietary recall, thus may not reflect usual intake”
Comment 7: Consider the NOVA system is a food classification based on the nature, extent, and purpose of industrial food processing but the food composition database is not designed to categorise foods in this way so errors related to classification of foods cannot be excluded.
Response 7: We have now included in the limitations (page 6): “Furthermore, the NOVA classification categorizes foods according to food processing and not nutritional content. The food composition database is not designed to categorise foods in this way thus errors related to classification of foods cannot be excluded.”
Comment 8: Line 322 Please describe why it is problematic that >8000 foods have a health star rating?
Response 8: We considered this, and have changed ‘problematically’ to ‘Related…’…
Comment 9: Line 334 Yes agree a multi system approach is clearly needed. Furthermore UPF are a dominant component of the food supply and this raises environmental concerns as these come at the expense of the cultivation, manufacture and consumption of traditional minimally processed foods.
Response 9: Thankyou.
Reviewer 2 Report
It is a solid piece on the consumption of ultra-processed foods by Australian women. I read it with interest. I think the work has excellent potential and it only need some minor revisions. I have made some suggestions, I hope you find them helpful.
Abstract
L19 - I had difficulty understanding this sentence. Are you suggesting replacing 25% of the UPF with unprocessed foods?
L21 - 75% of the UPF?
After reading the paper is easier to understand those sentence. However I had trouble in the first time.
Why is increasing the amount of food important? This is unclear to me.
Introduction
L37 - Although I understand and agree with the sentence, it seems rather general. Some of this population has poor nutritional quality. Do men or older women have a better diet? I do not think so... Perhaps some fine-tuning is needed here.
I think it's worth noting that consumers have difficulty classifying UPF, categorising several of them as healthy
L57 - I found this sentence odd. What alcoholic beverage is considered UPF? As far as I know, alcoholic beverages are classified as processed foods. Only light alcoholic sweet drinks are UPF (in my country). I suggest to be clear here and prevent the reader from classifying beer and wine as UPF.
I think the authors need to clearly emphasise the novelty of the study at the end of the introduction. Why is this study relevant? What is new?
Methods
The method is very clear
Result
Table 1/ Table 2 - Is it possible to add the Kcal value?
Do not you think all these comparisons should be done by statistical analysis? How are you sure that the differences between the models are significant? Some differences are very noticeable (e.g. quantity), others are not (e.g. added sugar).
Figure 1 - As there are no restrictions on colour representation for Nutrients, I suggest adding some colour here (or clearer shades of grey/black).
Discussion
The discussion is great. However, I missed a discussion on food prices. Reducing UPF and increasing minimally processed food could increase food spending. How could we deal with this?
Author Response
Dear Editor and Reviewers,
We thankyou very much for the kind and constructive feedback that has been provided on our manuscript (nutrients-1672887). We have amended our manuscript using track changes, and we have made a point-by-point response to each of the comments below. We have also uploaded a graphical abstract.
Thankyou for considering our manuscript.
Jessica Grieger
Comment 1: It is a solid piece on the consumption of ultra-processed foods by Australian women. I read it with interest. I think the work has excellent potential and it only need some minor revisions. I have made some suggestions, I hope you find them helpful.
Response 1: Thankyou for your kind opinion of our paper.
Abstract
Comment 2: L19 - I had difficulty understanding this sentence. Are you suggesting replacing 25% of the UPF with unprocessed foods?
L21 - 75% of the UPF?
After reading the paper is easier to understand those sentence. However I had trouble in the first time.
Response 2: Thankyou, we have now clarified this in the abstract.
Comment 3: Why is increasing the amount of food important? This is unclear to me.
Response 3: We have removed this statement of the results, as it is not essential to the abstract.
Introduction
Comment 4: L37 - Although I understand and agree with the sentence, it seems rather general. Some of this population has poor nutritional quality. Do men or older women have a better diet? I do not think so... Perhaps some fine-tuning is needed here.
Response 4: We agree in that woman, and in this case, reproductive aged women, have poor nutritional quality, as reflected by the studies we have cited, and this can impact overall health across the life stages. We have recently published showing that older women of reproductive age do not have better diet quality than younger women.
Comment 5: I think it's worth noting that consumers have difficulty classifying UPF, categorising several of them as healthy
Comment 5: Thankyou, we have now included, page 1, “While most young and older consumers understand the term ‘ultra-processed’, and they can correctly classify items such as soft drinks, biscuits and confectionary, they tend to mis-classify some more healthful foods such as milk, flour, meat, cheese, and bread [17,18], potentially contributing to lower intakes of these foods.”
Comment 6: L57 - I found this sentence odd. What alcoholic beverage is considered UPF? As far as I know, alcoholic beverages are classified as processed foods. Only light alcoholic sweet drinks are UPF (in my country). I suggest to be clear here and prevent the reader from classifying beer and wine as UPF.
Response 6: Thankyou for this comment. Upon reflection, we have removed the sentences in the introduction on alcohol.
Comment 7: I think the authors need to clearly emphasise the novelty of the study at the end of the introduction. Why is this study relevant? What is new?
Response 7: We have added to the introduction: “No studies have assessed the impact of theoretical changes to UPF consumption using Australian data. This is clearly of relevance, particularly in women of reproductive age given their overall poor dietary quality, increasing rates of overweight and obesity, and that diet and lifestyle intervention studies are currently ineffective…” “Outcomes from the modelling scenarios will enable the development of dietary interventions to improve diet quality and support body weight loss in these at-risk women.”
Methods
Comment 8: The method is very clear
Response 8: Thankyou.
Result
Comment 9: Table 1/ Table 2 - Is it possible to add the Kcal value?
Response 9: We have added the Kcal values to Table 2 and throughout the text and have included a footnote for the conversion in Table 1.
Comment 10: Do not you think all these comparisons should be done by statistical analysis? How are you sure that the differences between the models are significant? Some differences are very noticeable (e.g. quantity), others are not (e.g. added sugar).
Response 10: Nutritional modelling is about estimated and theoretical impacts so the use of statistical analyses comparing models is not needed.
Comment 11: Figure 1 - As there are no restrictions on colour representation for Nutrients, I suggest adding some colour here (or clearer shades of grey/black).
Response 11: We have now added colour to the figures.
Discussion
Comment 12: The discussion is great. However, I missed a discussion on food prices. Reducing UPF and increasing minimally processed food could increase food spending. How could we deal with this?
Response 12: We acknowledge that replacing UPF with unprocessed or minimally processed foods may impact the family food cost; however, unfortunately we don’t have the cost of UPF vs non UPF foods in Australia. We have made a comment on this as well as household income and UPF consumption in Australia. ““Interestingly, Australian data show that the lowest household income quintile consumes less UPF [65]. However, this may reflect healthy diets being cheaper than non-healthy diets in general in Australia [66,67]; the cost of UPF vs non-UPF is not currently available. Thus while the proposed dietary scenarios may impact the family food cost, future research on the potential impact on food prices will be helpful.”